# A Comparison of Phenolic, Flavonoid, and Amino Acid Compositions and In Vitro Antioxidant and Neuroprotective Activities in Thai Plant Protein Extracts

**DOI:** 10.3390/molecules29132990

**Published:** 2024-06-23

**Authors:** Pontapan Polyiam, Wipawee Thukhammee

**Affiliations:** 1Department of Physiology, Graduate School (Neuroscience Program), Faculty of Medicine, Khon Kaen University, Khon Kaen 40002, Thailand; pontpo@kku.ac.th; 2Human High Performance and Health Promotion (HHP&HP) Research Institute, Khon Kaen University, Khon Kaen 40002, Thailand; 3Department of Physiology, Faculty of Medicine, Khon Kaen University, Khon Kaen 40002, Thailand

**Keywords:** antioxidants, neuroprotection, phytochemical content, plant protein extracts

## Abstract

The leaves of mulberry, *Azolla* spp., sunflower sprouts, cashew nut, and mung bean are considered rich sources of plant protein with high levels of branched-chain amino acids. Furthermore, they contain beneficial phytochemicals such as antioxidants and anti-inflammatory agents. Additionally, there are reports suggesting that an adequate consumption of amino acids can reduce nerve cell damage, delay the onset of memory impairment, and improve sleep quality. In this study, protein isolates were prepared from the leaves of mulberry, *Azolla* spp., sunflower sprouts, cashew nut, and mung bean. The amino acid profile, dietary fiber content, phenolic content, and flavonoid content were evaluated. Pharmacological properties, such as antioxidant, anticholinesterase, monoamine oxidase, and γ-aminobutyric acid transaminase (GABA-T) activities, were also assessed. This study found that concentrated protein from mung beans has a higher quantity of essential amino acids (52,161 mg/100 g protein) compared to concentrated protein from sunflower sprouts (47,386 mg/100 g protein), *Azolla* spp. (42,097 mg/100 g protein), cashew nut (26,710 mg/100 g protein), and mulberry leaves (8931 mg/100 g protein). The dietary fiber content ranged from 0.90% to 3.24%, while the phenolic content and flavonoid content ranged from 0.25 to 2.29 mg/g and 0.01 to 2.01 mg/g of sample, respectively. Sunflower sprout protein isolates exhibited the highest levels of dietary fiber (3.24%), phenolic content (2.292 ± 0.082 mg of GAE/g), and flavonoids (2.014 mg quercetin/g of sample). The biological efficacy evaluation found that concentrated protein extract from sunflower sprouts has the highest antioxidant activity; the percentages of inhibition of 1,1-diphenyl-2-picrylhydrazyl radical (DPPH) and 2,2′-azino-bis-(3-ethylbenzthiazoline-6-sulphonic acid) (ABTS) radical were 20.503 ± 0.288% and 18.496 ± 0.105%, respectively. Five plant-based proteins exhibited a potent inhibition of acetylcholinesterase (AChE) enzyme activity, monoamine oxidase (MAO) inhibition, and GABA-T ranging from 3.42% to 24.62%, 6.14% to 20.16%, and 2.03% to 21.99%, respectively. These findings suggest that these plant protein extracts can be used as natural resources for developing food supplements with neuroprotective activity.

## 1. Introduction

It may take several decades of development before plant-based proteins become one of the more popular and environmentally friendly products among consumers for wellness [1]. Antioxidants in plant-based proteins largely consist of phenolic compounds and vitamins. There have been numerous reports on the benefits of consuming essential amino acids from plants. Rich sources of polyphenolic compounds can help to prevent the development of numerous diseases, including neurological disorders such as stroke, Parkinson’s disease (PD), Alzheimer’s disease (AD), and mild cognitive impairment (MCI) [2]. In addition, some essential amino acids and phenolic compounds can also provide beneficial effects for brain functions by altering the homeostasis of neurotransmitters such as monoamine and acetylcholine via the suppression of monoamine oxidase (MAO) [3,4] and acetylcholinesterase (AChE), respectively [5].

*Azolla* species or Mosquito fern are potential sources of edible protein and thus promising nutraceuticals or ingredients in functional and health-promoting foods [6]. They contain a higher crude protein content and essential amino acid (EAA) composition (rich in lysine) than most green forage crops and other aquatic macrophytes [7]. They have shown good antioxidant activities of 1,1-diphenyl-2-picrylhydrazyl radical (DPPH)-radical [8]. Also, they significantly reduced the levels of lactate dehydrogenase (LDH), catalase (CAT), glutamate oxalate transaminase (GOT), glutamate pyruvate transaminase (GPT), and malondialdehyde (MDA), and significantly increased the levels of superoxide dismutase (SOD) and glutathione peroxidase (GSH-Px) in common carp fish [9]. A recent study on the subacute toxicity of *Azolla* species reported that a repeat-dose administration of the ethanolic extract of *A. microphylla* in Wistar rats showed that the extract was non-toxic, and a lethal dose (LD50) was more than 2000 mg/kg/day [10]. Although this is a common edible plant in Thailand, there are few studies that focus on the effects of *Azolla*-derived protein on humans.

Mulberry leaves contain high total phenolic content (TPC) and total flavonoid content (TFC), and their antioxidant activities were evaluated by determining the DPPH radical scavenging activity and 2,2′-azino-bis-(3-ethylbenzthiazoline-6-sulphonic acid (ABTS) radical cation scavenging capacity. The investigated features exhibit good nutritive and antioxidant attributes [11]. Powders of mulberry leaves have a significantly higher total crude protein content and total amino acids than silkworm droppings [12]. Mulberry leaves also have neuroprotective and cognition-enhancing effects, improving working memory and the suppression of acetylcholinesterase (AChE) and monoamine oxidase type A and type B (MAO-A, MAO-B) activities in Thai humans [13]. However, the functional protein extract of mulberry leaves is rarely reported on.

Mung bean (*Vigna radiata* L.) has been known to be a great source of protein and has high amounts of phytochemical compounds, including polyphenols, polysaccharides, and peptides, and it is becoming a popular functional food in promoting healthy eating [14]. Its antioxidant capacity was measured via DPPH and ABTS assay, which indicated positive correlations of ABTS free-radical scavenging capacity with total phenolic acids and total flavonoid contents [15]. Peptides obtained from the mung bean protein hydrolysate act as an angiotensin I-converting enzyme (ACE) inhibitor with an IC50 value of 0.64 mg protein/mL [16], which can be used for the prevention of cardiovascular disease. Our previous studies showed that mung bean protein treatment improved spatial memory in a Morris water maze test in bilateral ovariectomized (OVX) obese rats. The possible underlying mechanism might occur partly via the improvement in cholinergic function (AChE activity), oxidative stress status, and apoptosis [17]. Thus, with respect to the protein concentrate from mung bean, more data are needed in the study of in vitro neuroprotective activity in humans.

Cashew nut: The total amino acid composition of cashew nut-derived protein hydrolysate contained an abundance of essential amino acids, including lysine, leucine, phenylalanine, and histidine. Interestingly, it also contained a high amount of glutamic acid. Their concentrations were higher than the requirement pattern suggested by FAO/WHO (1990) for adult humans. Moreover, it also caused both antioxidant and anti-inflammatory activities in cerebral ischemic rats induced by the occlusion of the right middle cerebral artery (Rt.MCAO) [18]. However, the neuroprotective activity of cashew nut-derived protein concentrate has not been reported.

Sunflower sprouts are becoming increasingly popular throughout society for healthy eating. The interest in fresh, ready-to-eat microscale vegetables, particularly sprouted seeds and microgreens, has been on the rise in recent years around the globe [19]. Sunflower sprouts (*Helianthus annuus*) increased total phenolic and flavonoid levels, as well as the antioxidant activity compared to ungerminated seeds [20]. Biological activities and compounds of sunflower sprout revealed antioxidant effects, antimicrobial activity, antidiabetic effects, antihypertensive effects, anti-inflammatory activity, and wound healing effects [21], but data remain limited in the study of in vitro neuroprotective activity. A previous study indicated that the application of sucrose solution through spraying and soaking methods with a suitable concentration had the potential to improve antioxidant activity in sunflower sprout [22]. However, five plant-based protein concentrates did not demonstrate the comparison of antioxidant and neuroprotective activities as part of their potential for alternative plant-based protein supplementation. Therefore, this study aimed to determine the amino acid profiles, phytochemical contents (total phenolic content, total flavonoid content), antioxidant activities (DPPH and ABTS assay), and neuroprotective activities (Acetylcholinesterase (AChE) inhibition, Monoamine oxidase (MAO) inhibition, and γ-aminobutyric acid transaminase (GABA-T) inhibition) in Thai edible plant protein concentrates from cashew nut, mung-bean, mulberry leaves, *Azolla* spp., and sunflower sprouts.

## 2. Results

### 2.1. Amino Acid Profiles

The percentage yields of protein isolation were as follows: cashew nut, 10.89%; mulberry leaves, 8.30%; mung bean, 6.31%; *Azolla* spp., 9.30%; and sunflower sprouts, 10.20%. Essential amino acids (EAA), e.g., valine, leucine, and isoleucine, are general terms for Branched Chain Amino Acids (BCAA), which are metabolized by the body and used as sources of muscle energy. The results found that concentrated protein from mung beans has a higher quantity of essential amino acids compared to concentrated protein from *Azolla* spp., cashew nut, mulberry leaves, and sunflower sprouts, and higher than soy protein, which is a well-known beneficial protein source. Moreover, a large volume of BCAA was mostly found in *Azolla* spp. when compared with soy protein, as shown in Table 1.

### 2.2. Dietary Fiber, Total Phenolic Content (TPC), and Total Flavonoid Content (TFC)

The dietary fiber content ranged from 0.90% to 3.24%, while the phenolic content and flavonoid content ranged from 0.25 to 2.29 mg GAE/g and 0.009 to 2.014 mg Quercetin/g of sample, respectively. Sunflower sprout protein isolates exhibited the highest levels of dietary fiber, phenolic content, and flavonoids, significantly higher than cashew nuts, mung beans, mulberry leaves, and *Azolla* spp. (*p* < 0.001). It was found that among five types of plant-based protein concentrates, sunflower sprouts showed the highest total phenolic compounds at a concentration of 2.292 ± 0.082 mg of GAE/g. Cashew nut contained phenolic compound contents at a concentration of 1.117 ± 0.017 mg of GAE/g, while mulberry leaves, mung bean and *Azolla* spp. obtained phenolic compound contents at concentrations of 0.489 ± 0.006, 0.455 ± 0.006, and 0.246 ± 0.011 mg of GAE/g, respectively, as shown in Figure 1. The flavonoid contents of five types of plant-based protein concentrates showed that the sunflower sprouts contained the highest flavonoid content when expressed as mg of Quercetin/g. Cashew nut contained the lowest flavonoid content, but mulberry leaves showed a higher flavonoid content than mung-bean and *Azolla* spp., as shown in Figure 1.

### 2.3. Antioxidant Activity

Free radicals are involved in many disorders, including neurodegenerative diseases such as Alzheimer’s disease and Parkinson’s disease [23]. Antioxidants, through their scavenging power, are useful for the management of these diseases. The DPPH stable free radical method is an easy, rapid, and sensitive way to survey the antioxidant activity of the plant extracts [24]. At the mentioned concentration, sunflower sprouts showed the highest % inhibition of DPPH free radical scavenging ability, followed by *Azolla* spp., mung bean, mulberry leaves, and cashew nut (*p* < 0.001). ABTS inhibition ranging from 16.792 ± 0.094 to 28.333 ± 0.241 was observed. It was found that, in the comparison between the plant-based protein concentrates, sunflower sprouts showed the lowest IC50, followed by *Azolla* spp., mung-bean, cashew nut, and mulberry leaves respectively (*p* < 0.001). as shown in Figure 2.

### 2.4. Neuroprotective Activity

Accumulative lines of evidence have demonstrated that plants that show AChEI activity could enhance memory [25]. Monoamine oxidase (MAO) has been reported to play a crucial role in the examination of monoamine transmitters, including serotonin and the catecholamines dopamine, adrenaline, and noradrenaline [26]. All monoamine neurotransmitters are derived from the aromatic amino acids phenylalanine, tyrosine, and tryptophan. Recent findings have demonstrated the results of drug targeting in the suppression of MAO activity. It has been reported that numerous phytochemical ingredients also possess monoamine oxidase suppression activity [27]. Gamma-aminobutyric acid (GABA) is the most important inhibitory neurotransmitter in the human brain. GABA is rapidly converted to succinate and glutamate by GABA transaminase (GABA-T). There are numerous studies that have suggested that GABA plays a crucial role in calming the nervous system and promoting relaxation, which can help reduce anxiety and improve sleep quality.

Neuroprotective activities have been demonstrated in acetylcholinesterase (AChE) inhibition and monoamine oxidase (MAO) inhibition, and the γ-aminobutyric acid transaminase (GABA-T) inhibition of plant-based protein concentrates from cashew nut, mung bean, mulberry leaves, *Azolla* spp., and sunflower sprouts. Results found that protein from *Azolla* spp. exhibits the most potent inhibition of AChE enzyme, MAO, and GABA-T enzyme activities, followed by mulberry leaves, mung bean, and sunflower sprouts, respectively (*p* < 0.001), as shown in Figure 3.

## 3. Discussion

This study found that concentrated protein from mung bean has a high quantity of total essential amino acids (EAAs) of up to 52.16 g/100 g protein. According to a previous study, Tarahi M et al., 2024, demonstrated that EAAs (essential amino acids) of protein isolates from mung bean, whey, and soy were 43.60, 47.33, and 45.8 g/100 g protein, respectively [28]. This was higher than the FAO/WHO-recommended 32.8 g/100 g protein [29], suggesting that protein from mung bean is a protein source as good as soy and whey protein. The BCAAs’ recommended intake, for leucine, valine, and isoleucine, is 40, 20, and 19 mg/kg/day, respectively [30]. Additionally, it was found that mung bean protein contains high levels of phenylalanine. Phenylalanine is a precursor to the amino acid tyrosine and the neurotransmitters dopamine and norepinephrine, which play a crucial role in neuromodulation, controlling brain states, vigilance, action, reward, learning, and memory processes. Moreover, it was found that mung bean protein contains high levels of tryptophan, tyrosine, and glycine; it has been reported that tryptophan, tyrosine, and glycine are significantly restored in the brain, and the choline system is also improved by mung bean supplementation, which could upregulate the expression of brain-derived neurotrophic factor, postsynaptic density 95 protein (PSD95), synaptosome-associated protein 25 (SNAP25), downregulated Toll-like receptor 4 (TLR4), and nuclear factor kB (NF-kB). Metabolites in the serum also underwent changes. These suggested that malnutrition hinders neurodevelopment, while mung bean protein diet reversed this trend [31]. Plant-based proteins not only contain beneficial amino acids but also contain phytochemicals that are beneficial to the body. Moreover, glutamic acid is a non-essential amino acid found in a large volume in mung beans. Glutamic acid in the body plays an important role in cognition and mood regulation [32]. A previous study described the side-chain–backbone of N-acetyl-L-glutamine-N-methylamide as a potential energy surface. Glutamine has a similar structure to asparagine, but glutamine has two CH_2_ groups in its side-chain while asparagine has only one. It has been suggested that the glutamine side-chain can reach out further than that of asparagine. Thus, glutamine in a protein has not only a structural but a functional role to play as well [33]. Previous studies have found that mung bean protein contains vitexin, isovitexin, sinapic acid, and ferulic acid, which might be the major bioactive compounds for mung bean-mediated neuroprotection. The mechanisms involved include the inhibition of β-amyloidogenesis, the inhibition of tau hyperphosphorylation, a reduction in oxidative stress, a reduction in neuroinflammation, the promotion of autophagy, and the stimulation of AChE activity [34]. Moreover, it was found that mung bean has been documented to ameliorate hyperglycemia, hyperlipemia, and hypertension, as well as to possess hepatoprotective and immunomodulatory activities [14].

From the study data, it was found that protein concentrated from sunflower sprouts exhibited antioxidant properties, as evidenced by the highest DPPH and FRAP values. Free radicals are involved in many disorders, including neurodegenerative diseases such as Alzheimer’s disease and Parkinson’s disease [23]. Antioxidants, through their scavenging power, are useful for the management of those diseases. The protein concentrated from sunflower sprouts contains high levels of phenolic and flavonoids, which have been reported to have antioxidant properties [35]. Flavonoids are glycosides of flavones, and –OH and –OCH3 functionality may be focused around the three rings of flavones. This structure of the pyranone ring may also occur in its hydrogenated or reduced form, and indicates that the reduction of the pyranone ring may also be caused by oxidation via hydrogenation. For this the reason, flavonoids have antioxidant activity, which is due to their ability to reduce free radical formation and to scavenge free radicals [36]. Furthermore, the concentrated protein from sunflower sprouts also contains a significant amount of BCAAs (leucine, isoleucine, and valine), which have been reported to have antioxidant effects [37]. Additionally, it was found that there were high levels of glycine and aromatic amino acids (AAAs), including phenylalanine, tryptophan, and tyrosine, which play a role in anti-inflammatory effects and improve neurotransmitter communication [38].

Proteins from *Azolla* spp. have the potential to inhibit the activity of enzymes such as AChE, MAO, and GABA-T. This could be due to the concentrated protein extracted from *Azolla* spp. containing significant amounts of tryptophan, which serves as a precursor to serotonin, playing a role in regulating mood, emotions, appetite, and sleep, and reducing symptoms of depression. Additionally, it contains melatonin precursors, which are involved in sleep regulation. Furthermore, it contains high levels of AAAs, which are monoamine neurotransmitter precursors involved in mood regulation and neurological disorders such as depression, anxiety, and Parkinson’s disease. Additionally, it was found that protein from sunflower sprouts and *Azolla* spp. contains high levels of dietary fiber. Dietary fiber can enhance the activity of beneficial gut microbes such as *Lactobacillus* and *Bifidobacterium*, resulting in increased levels of short-chain fatty acids (SCFAs) [39]. Dietary fibers have significant effects on the gut microbiota composition, which affects the type and amount of SCFAs produced [40]. SCFAs are the metabolic substrates that can modulate cellular metabolism; lipid metabolism plays an essential role in modulating the integrity of the epithelial barrier, controlling the immune and inflame responses [41]. Furthermore, SCFAs might directly contribute to brain function by promoting blood–brain barrier (BBB) integrity, neurotransmission, and levels of brain-derived neurotrophic factor (BDNF), and influencing cognition. A study has indicated a role of SCFAs in the pathway of the gut–brain axis [42].

Cashew nut protein contains high levels of arginine. There have been reports on the benefits of arginine in regulating blood pressure and stimulating the immune system [43]. Additionally, a previous study found that the consumption of cashew nut-derived protein hydrolysate with high fiber showed a high potential for antioxidant and anti-inflammatory activities. Cashew nut-derived protein hydrolysate with high fiber significantly decreased brain infarction and oxidative stress in Middle Cerebral Artery Occlusion (MCAO) rats. Additionally, it was found to improve memory and reduce levels of cholesterol, triglycerides, and LDL [18]. Moreover, it was found that a cashew nut diet enhanced the inhibitory impact on activities of AChE and MAO in cisplatin-induced oxidative damage to brain rats, and also boosted redox equilibrium and exhibited protection by increasing SOD, CAT, GST, and GPx activities [44]. A study by Dias et al. [45] demonstrated that cashew nut, rich in fatty acids and phenolic and flavonoid compounds, reduced the anxiogenic-like behavior caused by dyslipidemia in rats without altering brain fatty acids.

There have been reports that mulberry leaves and their components possess antioxidant effects, reduce brain infarct volume after stroke, improve cognitive function and learning processes, and reduce memory impairment in various animal models. Mulberry and its extracts ameliorate Parkinson’s disease-like behaviors, limit the complications of diabetes mellitus on the central nervous system, and have anticonvulsant, antidepressant, and anxiolytic effects [46]. Moreover, it was found that mulberry leaves also represent neuroprotective and cognition-enhancing effects through improvements in working memory and the suppression of Acetylcholinesterase (AChE) and monoamine oxidase type A and type B (MAO-A, MAO-B) activities in Thai humans [13].

In conclusion, it is evident that mung bean protein and sunflower sprout protein contain high levels of essential amino acids and dietary fiber, as well as exhibiting strong antioxidant activity. Additionally, it was found that protein from *Azolla* spp. and mulberry leaves has good neuroprotective effects.

## 4. Materials and Methods

### 4.1. Plant Material and Extractions

Mung-bean and *Azolla* spp. came from the Office of Agricultural Research and Development Region 3, Khon Kaen province, Thailand. Mulberry leaves came from Queen Sirikit Sericulture Center Khon Kaen, Thailand. Cashew nut Sisaket 60-1 spp. and sunflower sprouts were purchased from a farmer in Khon Kaen province, Thailand.

Protein extraction modified from Ogunwolu et al. [47]’s method involved mixing 1 part sample with 10 parts distilled water. The protein was extracted under alkaline conditions by adjusting the pH of the solution to 9.0 using 2 N NaOH and stirring at 1000 rpm for 2 h using an IKA^®^ RW20 digital stirrer (Staufen, Germany). The solution was then centrifuged at 8000× *g* for 20 min at 4 degrees Celsius using a Sorvall^®^ RC 6 Plus centrifuge (North Carolina, USA). The supernatant was collected and adjusted to pH 4.5 with 2 N HCl to precipitate the protein. After stirring at 1000 rpm for 1 h, the solution was centrifuged again at 8000× *g* for 20 min at 4 degrees Celsius to obtain a concentrated protein pellet. The protein pellet was then mixed with distilled water in a ratio of 1:10 and the pH was adjusted to 7.0 with 2 N NaOH. Stirring at 1000 rpm for 1 h resulted in a concentrated protein solution. This solution was further centrifuged at 8000× *g* for 20 min, and then the clear supernatant was encapsulated by adding 5% maltodextrin to obtain a homogeneous mixture, preparing it for subsequent spray drying.

### 4.2. Identification of the Amino Acid Profile

The results were reported by the Laboratory, ALS Laboratory Group (Bangkok, Thailand), based on AOAC (2016), 994.12: 100 g of prepared spray-dried powder of each protein (at room temperature), was sent by post 2–3 days before analysis by the ALS Laboratory.

### 4.3. Determination of Dietary Fiber

The results were reported by the Testing Laboratory: Central Laboratory (Bangkok, Thailand) CO., LTD. Total dietary fiber in foods (enzymatic–gravimetric method) was based on AOAC (2019), 985.29: 100 g of prepared spray dried powder of each protein (at room temperature), was sent by post 2–3 days before analysis by the Central Laboratory (Bangkok, Thailand).

### 4.4. Determination of Total Phenolic Contents

Total phenolic contents in each of plant protein extract were determined using the modified Folin–Ciocalteu method [48]. Briefly, reactions were carried out in 96-well microtiter plates consisting of 20 µL of extract sample (10 mg/mL in methanol) mixed with 120 µL of Folin–Ciocalteu reagent (50% *v*/*v*), and incubated at room temperature for 8 min, avoiding the light. Then, 30 µL of 20% sodium carbonate (Na_2_CO_3_) was added and incubated at 30 ± 2 °C in the dark for 2 h. After the incubation, the absorbance of the plate was measured at 765 nm by a microplate reader. Various concentrations (0.5–500 µg/mL) of Gallic acid were used to prepare the standard calibration curve. The result was expressed as mg gallic acid equivalent (GAE)/g sample.

### 4.5. Determination of Total Flavonoid Content

The determination of the total flavonoid content (TFC) in each of the extract samples was carried out according to the method previously described [49]. Quercetin was used as the standard to make a standard curve. Stock quercetin solution was prepared by dissolving 0.1 mg quercetin in 1.0 mL ethanol, and then the standard solutions of quercetin were prepared by serial dilutions using ethanol (1–50 μg/mL). An amount of 100 μL diluted standard quercetin solution or extract was mixed separately with 50 μL of 2% aluminum chloride. After mixing, the solution was incubated for 60 min at room temperature. The absorbance of the reaction mixtures was measured against a blank at 415 nm wavelength with a microplate reader. The concentration of TFC in the rice sprout extract was extrapolated from the calibration plot of quercetin. The results were expressed as mg quercetin equivalent (QE)/g of sample. All the determinations were carried out in triplicate.

### 4.6. Determination of Antioxidant Activity

#### 4.6.1. The 1,1-Diphenyl-2-picryl hydrazyl Radical (DPPH) Assay

Briefly, 0.15 mM DPPH was aliquoted in 180 µL of methanol and added to 20 µL of the sample at various concentrations (5–1000 µg/mL), with incubation at 30 ± 2 °C, avoiding light, for 30 min. After incubation, the sample was measured with a microplate reader at an absorbance of 517 nm [50]. Ascorbic acid was used as a positive control. Results was reported as the IC50 (concentration in micrograms per milliliter required to inhibit DPPH radical formation by 50%) and %Inhibition of DPPH.

#### 4.6.2. The 2,2′-azinobis-3-ethylbenzothiazoline-6-sulfonic Acid (ABTS) Assay

An ABTS assay was performed to investigate the activity of free radical scavenging in each sample. ABTS^•^+ solution, a mixture of 2.45 mM potassium persulfate (K_2_S_2_O_8_) and 7 mM ABTS (3:2 *v*/*v*) and diluted with deionized water (1:20 *v*/*v*), was used as a working solution. A volume of 20 µL of the sample at various concentrations was mixed with 40 µL of distilled water and 150 µL of ABTS^•^+ solution and the absorbance at 734 nm was measured with a microplate reader [51]. Trolox was used as a standard. Results were reported as an IC50 value (concentration in micrograms per milliliter required to inhibit ABTS radical formation by 50%).

### 4.7. Determination of Neuroprotective Activity

#### 4.7.1. Acetylcholinesterase (AChE) Inhibition Assay

The AChE suppression activity of each extract was determined using the colorimetric method according to the method previously described [52]. Briefly, 25 µL of the sample at various concentrations was incubated with the reaction mixture containing 25 µL of 15 mM acetyl thiocholine iodide (ATCI), 75 µL of 3 mM 5,5′-dithio-bis-2-nitrobenzoic acid (DTNB), and 50 µL of 50 mM Tris-HCl (pH 8.0) at room temperature for 5 min. After mixing, the absorbance at 412 nm was recorded with a microplate reader both before and after adding 0.22 U/mL of AChE (25 µL). The percentage of inhibition was calculated by comparison between the rate of hydrolysis of the ATCI in the samples and that of the blank (Tris-buffer). Donepezil (ARICEPT^®^, NY, USA) was used as a reference standard. The AChE inhibition activity of each sample was expressed in terms of IC50. Each sample was assessed in triplicate.

#### 4.7.2. Monoamine Oxidase (MAO) Inhibition Assay

Total MAO is an important enzyme in the metabolism of a wide range of endogenous monoamine neurotransmitters, such as noradrenaline, dopamine, and serotonin (5-HT). MAO inhibition activity was measured by using the colorimetric method. The chromogenic solution that was prepared for inclusion in the assay mixture contained 1mM vanillic acid, 500 µM 4-aminoantipyrine, and peroxidase (4 U/mL) in potassium phosphate buffer (0.2 M, pH 7.6). The rat cerebral cortex tissue sample was isolated and prepared as a homogenate in 0.1 M potassium phosphate buffer, pH 7.4 (1:5 *w*/*v*) and subjected to 12,000 rpm centrifugation at 4 °C for 10 min. The supernatant was harvested and served as the source of monoamine oxidase. The assay mixture contained 25 µL of monoamine oxidase, 25 µL of various concentrations of extract, 50 µL of chromogenic solution, and 200 µL of 500 µM P-Tyramine (for Total MAO). After mixing, the reaction mixture was incubated at 37 °C for 30 min, and the absorbance at 490 nm was recorded with a microplate reader [53]. Results were expressed as IC50. Each sample was assessed in triplicate.

#### 4.7.3. γ-Aminobutyric Acid Transaminase (GABA-T) Inhibition

GABA-T activity was performed using an enzymatic assay, according to Jung et al. [54]. The method involves the conversion of GABA to succinic acid through the consecutive reactions of GABA-T (in the sample) and semialdehyde dehydrogenase (added in the assay). The rat cerebral cortex tissue sample was isolated and prepared as a homogenate in 0.1 M potassium phosphate buffer, pH 7.4 (1:5 *w*/*v*), and subjected to 12,000 rpm centrifugation at 4 °C for 10 min. The supernatant was harvested and served as the source of GABA-T. The assay mixture contained 800 µL of GABA-T Buffer (20 mM of Gamma-Aminobutyric acid (GABA), 10 mM of Alpha-ketoglutarate, and 0.5 Mm NAD in 0.5M sodium phosphate buffer, pH 8.0), 200 µL of brain supernatant, and 200 µL of plant protein sample. In the reactions NAD+ was reduced to NADH, allowing the quantification of GABA-T through spectrophotometric measuring at 340 nm. Results were expressed as IC50. Each sample was assessed in triplicate.

### 4.8. Statistical Analysis

Experimental results are presented as means ± standard error of the mean (SEM) except for the proximate tests, which were carried out in triplicate. Between-group statistical analyses were performed using a one–way ANOVA followed by Tukey’s multiple comparison tests. The significant difference was set at *p*-value < 0.05. Regression and partial correlation analyses were performed using SPSS 19 (SPSS Inc., Chicago, IL, USA).

## 5. Conclusions

This study represents the results of the findings on the amino acid profile and antioxidant and neuroprotective activities of five plant-based protein concentrates, including cashew nut, mung bean, mulberry leaves, *Azolla* spp., and sunflower sprouts. In conclusion, it is evident that mung bean protein and sunflower sprout protein contain high levels of essential amino acids and dietary fiber, as well as exhibiting strong antioxidant activity. Moreover, the concentrated protein-derived mung bean revealed the highest level of arginine, which can regulate synaptic plasticity and neurogenesis. Additionally, it was found that protein from *Azolla* spp. and mulberry leaves has good neuroprotective effects. This study shows that proteins from various plant sources have different health benefits. The proteins from all five plant types have the potential to be developed into functional ingredients and functional foods. However, further studies in animal models or with human volunteers are needed, including investigations into the mechanisms of action, such as molecular biology tests and immunohistochemistry tests, to determine the precise efficacy and mechanisms of action.

## Figures and Tables

**Figure 1 molecules-29-02990-f001:**
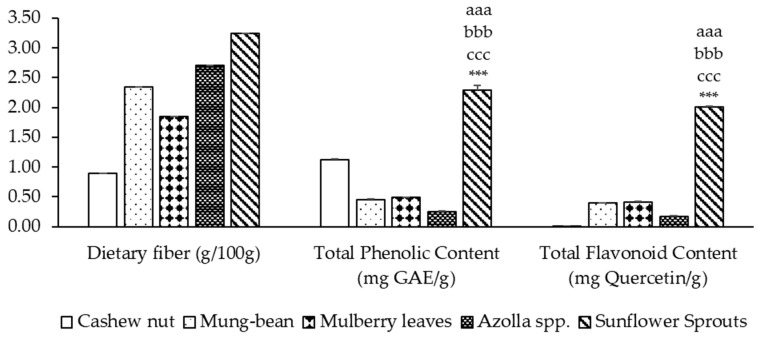
Dietary fiber and phytochemical contents of plant-based protein concentrates. Results of TPC and TFC are presented as the mean ± SEM (*n* = 3). ^aaa, bbb, ccc,^ *** *p*-value < 0.001 compare cashew nut, mung bean, mulberry leaves, and *Azolla* spp., respectively.

**Figure 2 molecules-29-02990-f002:**
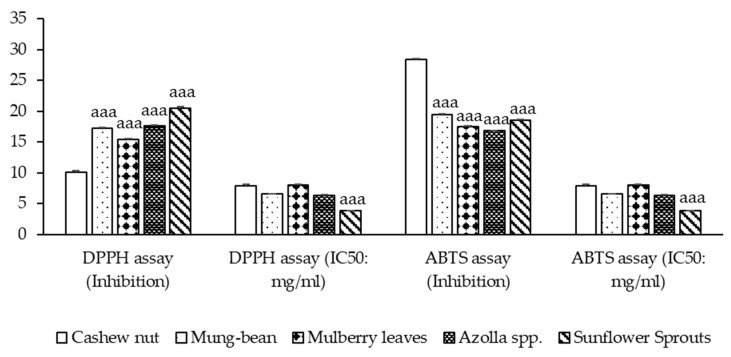
Antioxidant activity of plant-based protein concentrates and percentage presence of DPPH (1,1-diphenyl-2-picrylhydrazyl radical) and ABTS (2,2′-azino-bis-(3-ethylbenzthiazoline-6-sulphonic acid radical inhibition) and IC50 (Half maximal inhibitory concentration) are presented as the mean ± SEM (*n* = 3). ^aaa^
*p*-value < 0.001 compared to cashew nut.

**Figure 3 molecules-29-02990-f003:**
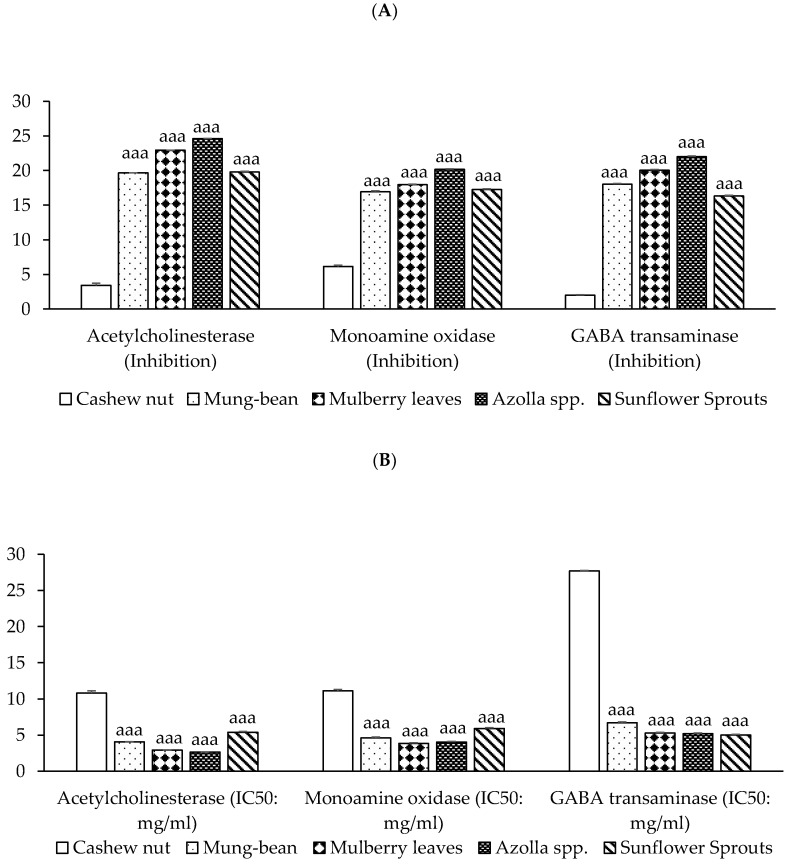
Neuroprotective activities of plant-based protein concentrates: (**A**) percentage of presence of acetylcholinesterase (AChE), monoamine oxidase (MAO), and GABA-T Inhibition and (**B**) IC50 (half maximal inhibitory concentration). Results of neuroprotective activities are presented as the mean ± SEM (*n* = 3). ^aaa^
*p*-value < 0.001 compared to cashew nut.

**Table 1 molecules-29-02990-t001:** Amino acid profiles of plant-based protein concentrate.

Amino Acid Profile	Protein Concentrate (mg/100 g Protein)					
	Soy Protein	Cashew Nut	Mung Bean	Mulberry Leaves	*Azolla* spp.	Sunflower Sprouts
Essential amino acid						
Threonine (Thr)	2474	1302	2733	715	4487	5026
Methionine (Met)	814	651	1068	221	2243	1400
Phenylalanine (Phe)	3278	2605	6110	800	4829	5358
Histidine (His)	1578	3257	7362	1345	1414	2464
Lysine (Lys)	3929	2931	6736	1030	5853	5334
Valine (Val)	3064	2280	4134	810	5121	6996
Isoleucine (Ile)	2942	1302	3170	565	3512	4793
Leucine (Leu)	4917	4560	8023	1425	8926	8640
Tryptophan (Trp)	835	ND	664	605	682	1540
Arginine (Arg)	4642	7817	12,155	1415	5024	5831
Total EAAs	28,473	26,710	52,161	8931	42,097	47,386
Non-essential amino acid						
Serine (Ser)	3369	2931	5434	675	5024	4625
Glycine (Gly)	2688	2280	1739	408	4829	5530
Glutamic acid (Glu)	12,013	12,377	15,269	1440	10,000	13,254
Proline (Pro)	3298	2931	4817	925	5658	4528
Cysteine (Cys)	886	1628	473	ND	1609	ND
Alanine (Ala)	2677	1628	3970	945	6390	6247
Tyrosine (Tyr)	2301	4234	2416	482	2829	3381
Aspartic acid	7249	1302	12,245	1735	8975	10,504
Total NEAAs	34,481	29,315	46,366	6611	45,316	48,073
HAA	24,513	18,241	33,700	6704	42,195	45,036
AAA	5579	6840	8526	1282	7658	10,281
TAA	62,954	56,026	98,527	15,542	87,414	95,460

HAA: Hydrophobic amino acids: glycine (Gly), alanine (Ala), valine (Val), leucine (Leu), isoleucine (Ile), proline (Pro), phenylalanine (Phe), methionine (Met), and tryptophan (Trp). AAA: Aromatic amino acids (Phe, Trp, and Tyr). TAA: Total amino acids; ND, not detected.

## Data Availability

Data are contained within the article.

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
