# Peer review of "A Comparison of Phenolic, Flavonoid, and Amino Acid Compositions and In Vitro Antioxidant and Neuroprotective Activities in Thai Plant Protein Extracts"

_molecules, 2024, doi:10.3390/molecules29132990_

Round 1
Reviewer 1 Report
Comments and Suggestions for Authors
-The keywords should be revised according to the journal writing rules and should be rearranged in alphabetical order
- Azolla and in vitro should be written in italics (including references and manuscript)
-The conclusion part should be rewritten, and a judgment should be made as a result of the data of this study.
-Table explanations should be presented in detail with their units, and the table should be clearly understandable when examined independently of the text.
-The discussion section is quite insufficient. The study results should be discussed in the context of previous studies, the literature is insufficient for discussion.
-unit usage should be standard, all text should be checked
-There are errors in the use of citations, the entire text should be checked
(Zhishen J et al., 1999) [38])
(Miller et 270 al., 1993) [40]. Jung et al., 1977 [43]. There are many that should be checked.
-All text should be rearranged according to the journal writing rules
Author Response
Thank you very much for the excellent and helpful suggestions. I have revised and improved the manuscript according to your recommendations. Please see the attachment.

Reviewer 2 Report
Comments and Suggestions for Authors
In this manuscript titled, “Comparison of Phenolic, Flavonoids, Amino Acid Composition, In Vitro Anti-oxidant and Neuroprotective Activities in Thai Plant Protein Extracts”, the authors aimed to determine amino acid profiles, phytochemical contents (Total phenolic content, Total flavonoid content), antioxidant activities (DPPH and ABTS assay), and neuroprotective activities (Acetylcholinesterase (AChE) inhibition and Monoamine oxidase (MAO) inhibition, and γ-aminobutyric acid transaminase (GABA-T) inhibition) in Thai edible plants protein concentrate from cashew nut, mung-bean, mulberry leaves, Azolla spp., and sunflower sprouts. The authors claim that the study presents that the protein extracts derived Thai indigenous edible plant including mulberry leaves, Azolla spp., sunflower sprouts, cashew nut and mung bean have antioxidant and neuroprotective activities. However, the quality of the manuscript and its results and conclusions, make the quality of the manuscript difficult to access, thus tempering my enthusiasm.
- Was any protein extracts derived from mulberry leaves, Azolla spp., sunflower sprouts, cashew nut and mung bean reported in any other disease models. Listing a few of them would be beneficial to the reader.
- The authors concluded that protein extracts from these indigenous species show antioxidant and neuroprotective activities by just doing biochemical estimations of amino acids, total phenolic content, DPPH assay, etc., however, performing some in-vitro or in-vivo studies would make the finding more robust.
- The authors claim to show neuroprotective potential of these protein extracts by MAO & AChE inhibition assay, however, doing some immunohistochemistry (like- Nissl Staining) on mice would further consolidate the finding of the manuscript.
Comments on the Quality of English Language
Moderate editing of English language is required.
Author Response
Thank you very much for the excellent and helpful suggestions. I have revised and improved the manuscript according to your recommendations.

Reviewer 3 Report
Comments and Suggestions for Authors
This study quantifies the amino acid profile, dietary fiber, phenolic and flavonoid content in protein isolates prepared from the leaves of mulberry, Azolla spp., sunflower sprouts, cashew nut and mung bean and related these to the pharmacological properties such as antioxidant, anticholinesterase, monoamine oxidase and γ-aminobutyric acid transaminase (GABA-T) activities.
The scope is adequate and suits the special issue “Functional Proteins in Food: Chemistry, Applications, and Health Benefits”; the methods are appropriate for the objectives.
However, the reproducibility is compromised by omitting the experimental details in section 4.2, 4.3. Are these new results or have they been reported before?
The manuscript needs to be edited better: English needs thorough revision because in the current state it affects clarity; Line106-114 should be a separate paragraph introducing the objectives; section 2.2 title does not reflect fiber content; reference style is not consistent, there are (Author, year) before [ref No.] in places, rather than consistent [ref No.]; one–way ANOVA followed by Tukey multiple comparison test results have not been presented thus can not make direct comparison amongst groups.
Result presentation is only boring tables. Can you add some plots to diversify?
References are limited. Suggest adding molecular simulation results exploring mechanisms: line 192, [31, 31a], 31a= An ab initio exploratory study of side chain conformations for selected backbone conformations of N-acetyl-L-glutamine-N-methylamide; line 199, [33, 33a], 33a= Ab initio and DFT conformational analysis of selected flavones: 5, 7-dihydroxyflavone (chrysin) and 7, 8-dihydroxyflavone.
Conclusion is overly brief please expand.
Comments on the Quality of English Languageextensive editing needed
Author Response
Thank you very much for the excellent and helpful suggestions. I have revised and improved the manuscript according to your recommendations. "Please see the attachment."

Round 2
Reviewer 1 Report
Comments and Suggestions for Authors
good work
Author Response
Thank you very much for all the advice.
Reviewer 2 Report
Comments and Suggestions for Authors
The manuscript titled “Comparison of Phenolic, Flavonoids, Amino Acid Composition, In Vitro Anti-oxidant and Neuroprotective Activities in Thai Plant Protein Extracts”, has been greatly revised and edited by the authors adding important portions in the introduction and results sections. The authors have included relevant figures depicting the dietary fiber, phytochemical contents, antioxidant and neuroprotective activity of the plant based protein concentrate in the revised manuscript. The authors have significantly improved their ‘Discussion’ section and added relevant references, making the revised manuscript more robust. However, I still have some specific comments that can be addressed:
· The authors claim that in their title, ‘In Vitro Anti-oxidant and Neuroprotective Activities’ and they have done specific experiments to claim the same, however, I am still curious to know which cell lines (since they claim it’s an in vitro study) did they use to evaluate the same. The details have not been provided in the ‘Materials and Methods’ sections.
· The quality of language can be improved to make it more palatable to the reader.
Comments on the Quality of English LanguageMinor editing of English language required.
Author Response
The authors claim that in their title, ‘In Vitro Anti-oxidant and Neuroprotective Activities’ and they have done specific experiments to claim the same, however, I am still curious to know which cell lines (since they claim it’s an in vitro study) did they use to evaluate the same. The details have not been provided in the ‘Materials and Methods’ sections.
ANSWER:
- DPPH : This method does not use cells DPPH assay is a method for analyzing antioxidant capacity which uses the reagent 2,2-diphenyl-1-picrylhydrazyl (DPPH) as a stable radical. This solution is purple in color. DPPH● will react with antioxidants. Receive hydrogen from antioxidants and get DPPH that has a lighter color or clear color.
- ABTS: This method does not use cell line. ABTS is a method for measuring the antioxidant capacity of substances. It is based on the reaction between antioxidants and positively charged free radicals of ABTS (ABTS•+). The ABTS assay measures the ability of antioxidants to bleach ABTS free radicals in the aqueous phase. Different ABTS•+ is created by the reaction between strong oxidizing agents and ABTS salts. ABTS•+ free radicals are then formed, which are blue-green colored substances.
In generally, Acetylcholinesterase (AChE), Monoamine oxidase (MAO), and γ-
aminobutyric acid transaminase (GABA-T) appear in neuronal cells and astrocytes in the brain tissue, it play role in neutralize neurotransmitters; Acetylcholine (ACh), Monoamine; 5-HT, dopamine or noradrenaline, and GABA, respectively, express during the neuron transmittion at synaptic cleft between pre-synaptic and post-synaptic neuron in the brain (the functions of the cholinergic and GABAergic systems.). Thus, polyphenols/amino acids properties inhibit AChE, MAO, and GABA-T leading to elevation of ACh, 5-HT, dopamine levels in the brain, as processing improve neuroprotection.
- AChE: This method does not use cells Acetyl thiocholine iodide when reacts with water with the enzyme acetylcholinesterase will become thiocholone. This substance reacts colorimetrically with 5,5´-dithio-bis-2-nitrobenzoic to produce a yellow compound.
- MAO and GABA-T : These two methods were tested using rat cerebral cortex tissue. I have added the identification in lines 362 and 373.
The quality of language can be improved to make it more palatable to the reader.
ANSWER: Thank you for the advice. I have consulted with the publisher regarding the language, and it is currently in progress."
Reviewer 3 Report
Comments and Suggestions for Authors
The majority of the issues raised have been addressed.
However, the newly added references [33] [36] are not in the revised manuscript. The revised manuscript must be in sync with "response to reviewer comments" file. Please make sure other points are also reflected in manuscript.
Comments on the Quality of English Language
minor check
Author Response
Thank you very much for all the advice.
However, the newly added references [33] [36] are not in the revised manuscript. The revised manuscript must be in sync with "response to reviewer comments" file. Please make sure other points are also reflected in manuscript.
Answer: Thank you for the advice.
references [33]: I've added the information in lines 213-217; 489-491.
references [36]: I've added the information in lines 231-234; 501-502.
[33] Xu, H.; Zhou, Q.; Liu, B.; Cheng, K.W.; Chen, F.; Wang, M.; Neuroprotective Potential of Mung Bean (Vigna radiata L.) Polyphenols in Alzheimer's Disease: A Review. J. Agric. Food Chem. 2021, 69, 11554-11571.
[36] Ullah, R.; Jo, M.H.; Riaz, M.; Alam, S.I.; Saeed, K.; Ali, W.; Rehman, I.U.; Ikram, M.; Kim, M.O. Glycine, the Smallest Amino Acid, Confers Neuroprotection against d-Galactose-Induced Neurodegeneration and Memory Impairment by Regulating c-Jun N-Terminal Kinase in the Mouse Brain. J. Neuroinflammation 2020, 17, 303.